*Version of October 26, 2023*

# Paul J. Crutzen – a pioneer in Earth system science and a founding member of the journal "Atmospheric Chemistry and Physics"

Rolf Müller[1], Ulrich Pöschl[2], Thomas Koop[3], Thomas Peter[4], and Ken Carslaw[5]

[1]Institute of Energy and Climate Research (IEK-7), Forschungszentrum Jülich, Jülich, Germany
[2]Max-Planck-Institut (MPI) für Chemie, Mainz, Germany
[3]Bielefeld University, Faculty of Chemistry, Universitätsstr. 25, Bielefeld, Germany
[4]ETH Zurich, Institute for Atmospheric and Climate Science, Zürich, Switzerland
[5]University of Leeds, Woodhouse Lane, Leeds, UK

**Correspondence:** Rolf Müller <ro.mueller@fz.juelich.de>

**Abstract.** Paul J. Crutzen was a pioneer in atmospheric sciences. At the same time, he was a kind-hearted, humorous person with empathy for the private lives of his colleagues and students, but also with the highest scientific standards for himself and for others. He made fundamental scientific contributions to a wide range of scientific topics in all parts of the atmosphere, from the mesosphere to the stratosphere and to the troposphere near the ground. In particular, he was the first to describe the $NO_x$-driven ozone depletion cycle in the stratosphere, he was among the first to develop the idea of chemical formation of ozone in the troposphere, he provided key concepts to explain the "ozone hole", and he made fundamental discoveries about the effects of biomass burning on the troposphere. Understanding and addressing the causes of man-made air pollution and climate change was the driving motivation for his scientific work. In his work he did not shy away from challenge and provocation. His work on smoke from fires after a potential nuclear war inspired new research on a concept now known as "nuclear winter". He also initiated the reopening of the debate on "geoengineering" – a concept now referred to as "climate intervention". He also brought the term "Anthropocene" to the popular debate. Moreover, he had a strong influence on atmospheric science through his educational role; there is a very large number of outstanding scientists, who started their career with scientific work together with Paul. In 2000, Paul was among the founders of the journal "Atmospheric Chemistry and Physics", which was unique at the time in providing public discussion of published preprints, and also what we now call "open access" to published articles. Paul's work on human impacts on atmosphere and climate has had a profound impact on the environmental policies of many countries for decades. In the future, his work will continue to be a guide for generations of scientists and environmental policy makers to come.

## 1 Introduction

Paul Crutzen was always full of scientific ideas that he pursued and also shared generously with colleagues and students. He was also a very hard-working individual; when he focused on a particular scientific problem, he could forget the world around him. Despite all his concentration on science, he always had time for his family, and never forgot how important the private

lives of his colleagues and students were. You could always discuss with him events of the day, from political issues to the weather and sports.

Particularly impressive about Paul's scientific achievements is the range of different topics in atmospheric science to which he made fundamental contributions (Müller, 2022; Fishman et al., 2023); a short overview is given below in section 2.2. Paul's research interests included topics in the mesosphere, the stratosphere and the troposphere, with a particular emphasis on the issues of climate change and air quality (e.g., Fishman and Crutzen, 1978; Fishman et al., 1979a, b); in this context, the role of aerosol particles – including black carbon – became a focus of his work (e.g., Lelieveld et al., 2001; Ramanathan et al., 2001).

Moreover, he was involved in the first studies on the global effects of a thick smoke layer in the atmosphere produced by fires caused by a possible nuclear war (Crutzen and Birks, 1982; Birks and Crutzen, 1983). This work inspired research on "nuclear winter" starting in the mid-1980s (e.g., Turco et al., 1983; Aleksandrov and Stenchikov, 1983; Robock, 1984; Covey et al., 1984). Paul has also sparked a new debate by breaking the taboo regarding a possible cooling of the climate by increasing the earth's albedo through stratospheric sulphur injections (Crutzen, 2006). Finally, he popularised the term "Anthropocene" as the

epoch dating from the commencement of geologically significant human impact on the Earth's system (Crutzen, 2002; Crutzen and Steffen, 2003; Crutzen and Müller, 2019; Benner et al., 2021, see also section 4).

Paul was a key figure in establishing the journal "Atmospheric Chemistry and Physics" (ACP). ACP has been a pioneer in transparent peer review since it was founded in the year 2000 (Pöschl, 2004, 2012; Ervens et al., 2023). The special issue "20 years of Atmospheric Chemistry and Physics" of which this paper is a part, and which is celebrating more than 20 years of ACP,

contains two papers that are directly related to topics that Paul brought up. These two papers are on the global consequences of a possible nuclear exchange (nuclear winter, Robock et al., 2023) and on climate intervention (Visioni et al., 2023).

Paul Crutzen himself provided a very good description of his life and of his scientific work in his published Nobel lecture (Crutzen, 1996) on the occasion of the 1995 Nobel Prize in Chemistry, which he shared with Mario J. Molina, and F. Sherwood Rowland. Shorter biographical texts are also available (Möllers et al., 2015; Lelieveld, 2021; Moortgat et al., 2021; Rodhe,

2021; Solomon, 2021; Zalasiewicz et al., 2021; Zetzsch, 2021). There are also two more recent and detailed memoirs describing Paul's life and his scientific achievements (Müller, 2022; Fishman et al., 2023) [1]. Finally, there is also a book (Lax, 2018) on the recent history (1959-2000) of the Max Planck Institute (MPI) for Chemistry ("Otto-Hahn-Institut") in Mainz, where Paul had worked since 1980; this book also contains a wide range of information on Paul's work during this period.

## 2    Paul Crutzen: the person and the scientist

### 2.1    The person

Paul Jozef Crutzen was born in Amsterdam on 3 December 1933 and passed away in Mainz on 28 January 2021. He was the son of Anna Gurk and Jozef Crutzen. In Amsterdam, on 14 February 1958, he married Terttu Soininen; Paul and Terttu have two daughters, Ilona and Sylvia, and three grandchildren (Müller, 2022). There is no doubt that Paul was a very hardworking

---

[1]See also the information on the web: https://www.mpic.de/3864489/paul-crutzen.

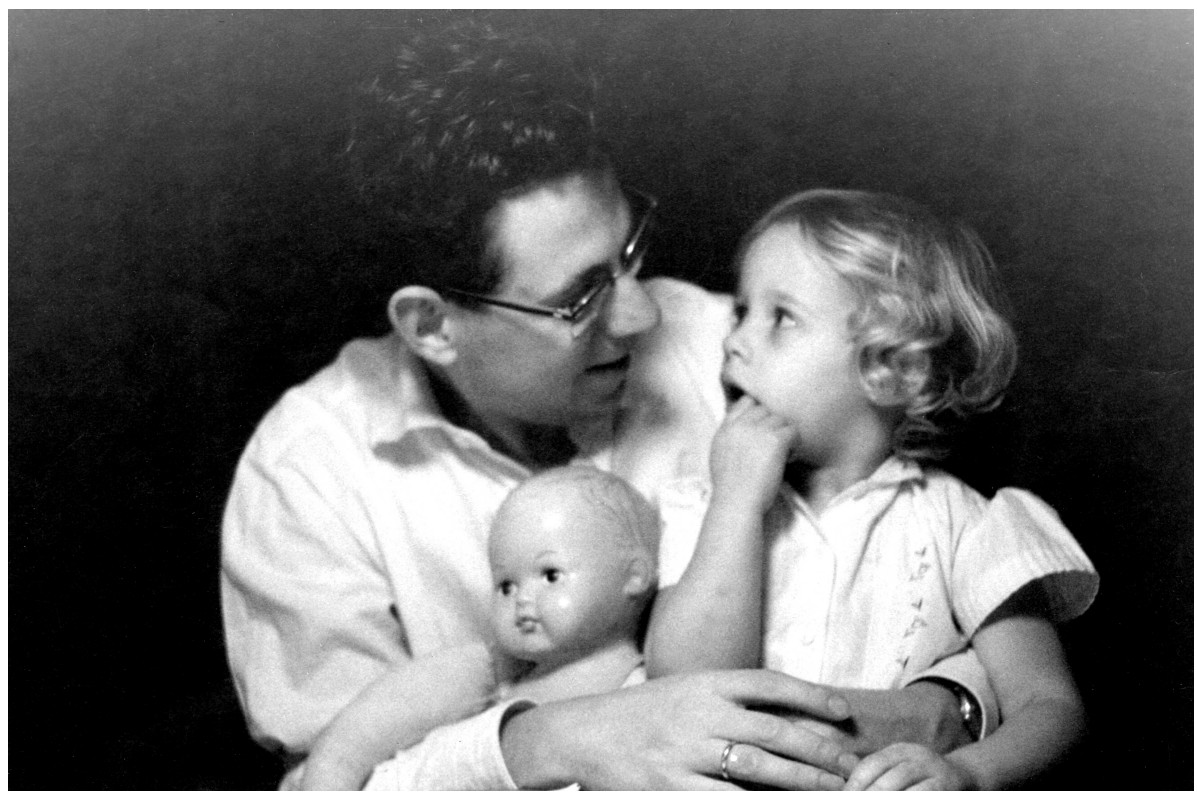

**Figure 1.** Paul Crutzen with his daughter Ilona in 1961. (Picture courtesy of Ilona Crutzen.)

man. He once said "see, this is the life of a scientist, always working". He was very dedicated and demanded the same of his
collaborators and students. If the only opportunity to talk science with him was on a Saturday afternoon, you had no choice but
to accept his invitation and come to the office. But this intensity also meant that he was always very interested in the work of
colleagues and students; you could count on a well-elaborated reply from Paul in a very short time frame to any scientific text
you sent him, be it a paper draft, parts of a doctoral thesis or any other kind of text.

However, no matter how much Paul concentrated on his scientific work, he always had time for his family (Fig. 1). He
himself mentioned that weekends were reserved for family, especially during the time when his daughters were young. The
entire Crutzen family has fond memories of family gatherings, weekends, holidays, and vacations with Paul. For his colleagues,
Paul was a very pleasant person to be with and science was not necessarily the main subject. One could chat with Paul over
a coffee or an evening meal on many other topics that he was interested in, such as sports. For example, he was always well-
informed about the performance of the Dutch ice skating team, or in football[2], particularly about the performance of the top
team of his hometown (Ajax Amsterdam). Moreover, he followed the football team of the town he lived in (1. FSV Mainz 05)
and watched matches live in the Mainz stadium.

---

[2]Football is the game referred to in American English as soccer.

## 2.2 The scientist

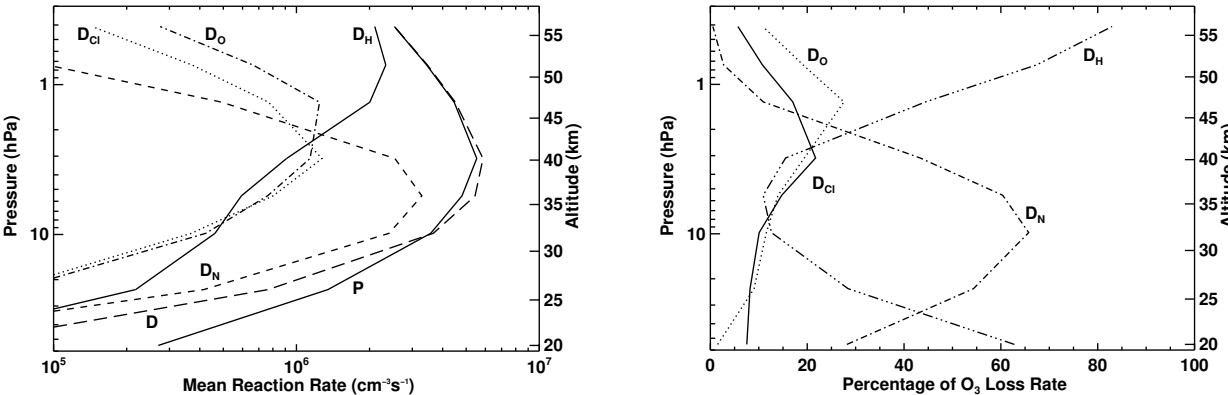

**Figure 2.** Dependence of various production and destruction reactions on altitude. Left hand panel: the mean reaction rates; right hand panel: the relative importance of the individual contributions to ozone loss in the gas-phase. $D_O$: Chapman reaction (the reaction $O_3 + O$), $D_N$: $NO_x$ catalysis (reactions 1 and 2) $D_H$: $HO_x$ catalysis (by H, OH, and $HO_2$), and $D_{Cl}$: $ClO_x$ catalysis (Molina and Rowland, 1974). P: production of odd oxygen (by the reaction $O_2 + h\nu$), D: total ozone destruction. The dominant ozone loss cycle in the stratosphere ($D_N$) was not known prior to Paul's work (Crutzen, 1970). (Figure adapted from Crutzen et al. (1995) and Grooß et al. (1999); figure courtesy of Jens-Uwe Grooß.)

Paul's scientific achievements are too numerous and their scope too broad to be covered in detail in this brief note (for more information, see e.g., Crutzen, 1996; Müller, 2022; Fishman et al., 2023). However, stratospheric ozone chemistry in general and $NO_x$ chemistry in particular were what started his scientific work even before his doctoral research (Müller, 2022). He proposed the groundbreaking idea that reactions catalysed by NO and $NO_2$ control the ozone concentration in the middle stratosphere (Crutzen, 1970), according to the following catalytic cycle:

$$NO + O_3 \quad \rightarrow \quad NO_2 + O_2 \tag{1}$$

$$NO_2 + O \quad \rightarrow \quad NO + O_2 \tag{2}$$

where the sum of NO and $NO_2$ is defined as $NO_x = NO + NO_2$ and O indicates an oxygen atom in ground-state ($O(^3P)$).

The discovery of this mechanism in 1970 was a crucial step towards facilitating a quantitative description of the chemistry of the stratospheric ozone layer; prior to Paul's discovery, the dominant catalytic loss cycle of stratospheric ozone ($D_N$, through reactions 1 and 2) was not known (Fig. 2). It is now known that $D_O$: the reaction $O_3 + O \rightarrow 2O_2$, originally proposed by Chapman (1930), is only a minor sink of stratospheric ozone and that $HO_x$-induced catalysis is important only close to the tropopause and above about 45 km (e.g., Portmann et al., 2012, see also Fig. 2). The recognition that chlorine ($D_{Cl}$) also catalytically contributes to stratospheric ozone loss came a few years after the discovery of the $NO_x$-induced cycle (Molina and Rowland, 1974).

The recognition of the strong effect of $NO_x$ on stratospheric ozone had a major impact, since emissions of $NO_x$ caused by a possible fleet of supersonic planes was the first stratospheric ozone depletion issue that was studied (Johnston, 1971; Crutzen, 1972). Paul also investigated how tropospheric nitrogen containing compounds (like $N_2O$) can enter the stratosphere and cause the formation of stratospheric $NO_x$ (Crutzen and Ehhalt, 1977; Müller, 2021). He initiated the first studies on the budget of $N_2O$ in the atmosphere and how it is influenced by human activity (Crutzen and Ehhalt, 1977).

Tropospheric chemistry, and in particular the chemical production of ozone in the troposphere were of great importance to Paul (Crutzen, 1996; Fishman et al., 2023). The radical OH (Levy, 1971; Crutzen and Zimmermann, 1991; Crutzen, 1996) is responsible for the oxidation of $CH_4$ (and many other compounds emitted into the atmosphere). It was found that in environments containing sufficient NO, the methane oxidation chain could produce ozone in large quantities – compared to the downward flux of ozone from the stratosphere to the troposphere – (Crutzen, 1973; Chameides and Walker, 1973). At that time, very little was known about the homogeneous and heterogeneous reactions affecting the methane oxidation chain in the troposphere, so that initial conclusions remained uncertain (Crutzen, 1974b). However some years later, together with Jack Fishman and Susan Solomon, Paul presented observational evidence for a strong *in situ* tropospheric ozone production (Fishman and Crutzen, 1978; Fishman et al., 1979b). Later, Paul's work on tropospheric ozone led him to realise the importance of biomass-burning for the chemistry of the atmosphere (Crutzen et al., 1979; Crutzen and Andreae, 1990).

Paul's work on smoke from fires after a possible nuclear war and the absorption of sunlight by the smoke (Crutzen and Birks, 1982; Birks and Crutzen, 1983) introduced the concept that the use of nuclear weapons would have global impacts that go much beyond the more obvious direct effects. This work inspired substantial research activity – Turco et al. (1983) and Aleksandrov and Stenchikov (1983), soon followed by Robock (1984) and Covey et al. (1984) calculated the surface temperature response to fires after a nuclear war and introduced the term "nuclear winter" (see also Robock et al., 2023). Paul's original intention, however, at the outset of these studies (Fishman et al., 2023) was to explore the impact on stratospheric ozone of nitrogen oxides that might form as a result of a possible nuclear war, a concept that had been discussed earlier (e.g., Whitten et al., 1975). Paul counted his contribution to this field as important from a political point of view. Indeed together with John Birks, Jeannie Peterson, Alan Robock, Carl Sagan, Georgiy Stenchikov, Brian Toon and Richard Turco he was presented with the 2022 "Future of Life Award" [3] by the "Future of Life Institute". The award was presented to this team for reducing the risk of nuclear war by developing and popularising the science of nuclear winter.

After the impact of chlorofluorocarbons (CFCs) on stratospheric ozone was identified (Molina and Rowland, 1974), Paul published a modelling study on this topic in the same year (Crutzen, 1974a). In 1985 the ozone hole was discovered by Farman et al. (1985). A year later, Paul – together with Frank Arnold – showed that the formation of stratospheric particles (well above the temperature threshold for ice formation) and the nitric acid uptake into these particles is a crucial aspect of ozone hole chemistry (Crutzen and Arnold, 1986). Heterogeneous chemistry (Solomon et al., 1986) and an ozone loss cycle specific for ozone hole conditions (Molina and Molina, 1987) turned out to be further key processes for explaining the chemical processes responsible for the formation of the ozone hole. Furthermore, Paul suggested that carbonyl sulfide (COS) constitutes the major non-volcanic source for aqueous sulphuric acid aerosol particles in the stratosphere (Crutzen, 1976).

---

[3]https://futureoflife.org/project/future-of-life-award/

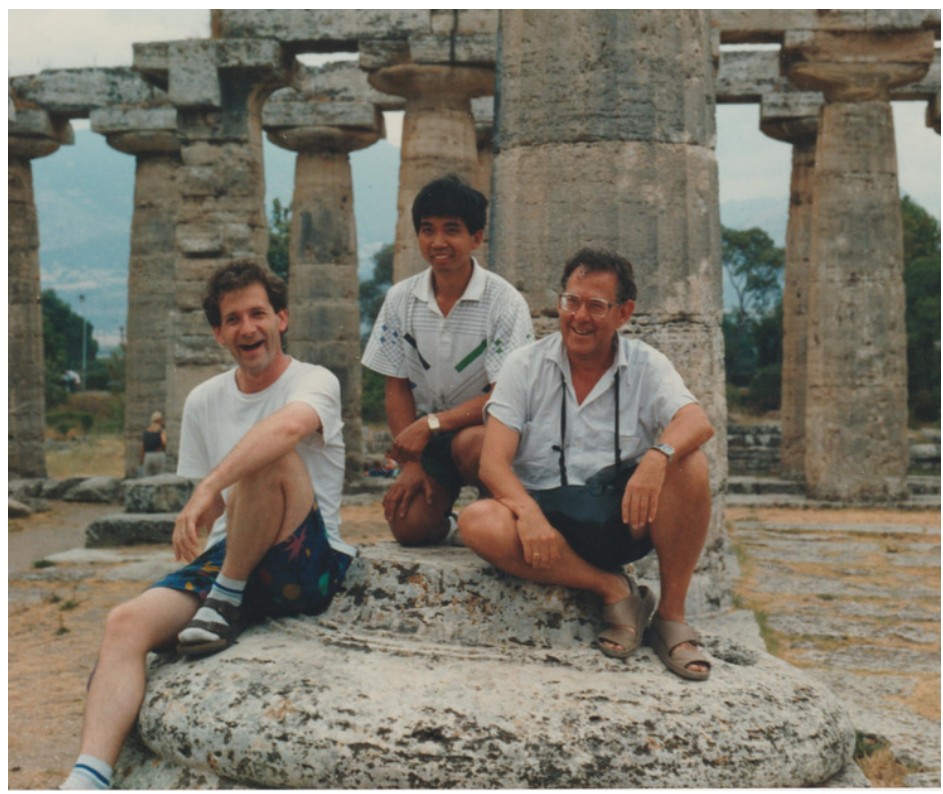

**Figure 3.** Paul Crutzen (right) during a summer school in 1993 together with Thomas Peter (left) and Beiping Luo (middle) at Paestum, Italy. (Picture by Thomas Koop.)

110      In recognition of the importance of the multi-phase chemistry on atmospheric aerosol particles (Abbatt and Ravishankara, 2023) and the many unknown processes regarding their microphysics, Paul initiated in the early nineties in Mainz a junior research group. The name of the group was: "Heterogene Chemie und Mikrophysik atmosphärischer Aerosolteilchen" [4]. An important starting point of this research was the paper by Luo et al. (1994) reporting on homogeneous and heterogeneous freezing rates of sulphuric acid droplets under stratospheric conditions, which has implications for the theory of the formation

115   of nitric acid trihydrate particles in the polar stratosphere. The three authors of that paper are shown in Fig. 3. This photograph was taken by Thomas Koop. At the same time the scene was photographed by A. R. Ravishankara, who also wanted to take a picture of the authors of the paper by Luo et al. (1994).

     Further work of the junior research group showed the occurrence and importance of a new type of polar stratospheric cloud (type Ib) that consisted of liquid rather than crystalline particles (Carslaw et al., 1993, 1994; Koop et al., 1995); other groups

120   also investigated liquid polar stratospheric clouds (Tabazadeh et al., 1994). It later became clear that these liquid clouds, and not the frozen nitric acid hydrate or ice particles, are the main hosts of heterogeneous chemical reactions responsible for chlorine

---

[4]in English: "Heterogeneous chemistry and microphysics of atmospheric aerosol particles".

activation and, thus, polar ozone depletion (e.g., Solomon, 1999; Kirner et al., 2015). In another study (Meilinger et al., 1995), the junior research group showed that the composition and freezing behaviour of the liquid particles depends on small-scale temperature fluctuations in the atmosphere. As it turned out, the smallest droplets reached higher $HNO_3$ concentrations than larger ones, thus unintuitively increasing the smaller droplets' likelihood to crystallise. When Paul, who was himself rather short in stature, first learned about this result during a discussion, he commented on it with a pinch of self-irony and a big smile: "Never neglect the small ones!".

A discussion about Paul as a scientist is not complete without talking about his educational role. Throughout his career, Paul interacted and collaborated with many important, influential scientists. These collaborations are easily noticeable by investigating his list of publications. But perhaps even more impressive is the educational impact he had. There is a very large number of outstanding scientists, who had – and have – an extraordinary career that started with a PhD, a postdoc or an early scientific interaction with Paul. The inspiration for these careers clearly came from Paul and they started at the various institutions at which he was active.

## 3   Paul Crutzen and the birth of a new journal

For many years (Ehhalt and Ridley, 1996), Paul Crutzen has been an editor of classical journals (*Tellus* and the *Journal of Atmospheric Chemistry*). In 2000, however, Paul was also key in the pioneering work of establishing a new, unique concept of scientific publishing: public discussion of preprints and open access publishing. Together with Uli Pöschl and Arne Richter, Paul helped founding the journal "Atmospheric Chemistry and Physics (ACP)", published by the European Geophysical Society (EGS, which is now the European Geosciences Union (EGU)). The first paper was submitted to ACP in 2001.

Scientific discussions have been documented before the invention of ACP; for example in the "Electronic Transactions in Artificial Intelligence" (ETAI) and the "Journal of Interactive Media in Education" (JIME) (Pöschl, 2012, section "Comparison to earlier initiatives with two- or multi-stage open peer review"). Further, the review process in "Faraday Transactions" and in the "Proceedings of the Combustion Institute" is "open", in the sense that these journals have a long history of meetings and their subsequent publication of the discussion. Faraday Discussions collects questions and answers through delegate discussion during meetings (rather than online or through a text forum); a discussion which is then edited and published alongside the articles in each volume. A similar procedure is followed by the Combustion Institute (see Nicovich and Ravishankara, 1982, for an example). The interactive open access process (as we call it today) with a multi-stage public peer review as practised in ACP, however, had not been introduced in scientific publishing prior to the launch of ACP (Pöschl, 2012).

Initially, when the concept of public public peer-review and a public discussion of submitted manuscripts was introduced, there was some confusion in the community about the status of discussion papers. Since then, however, it has become clear that discussion papers are preprints similar to the manuscripts posted on other preprint servers like arXiv.org but with the additional feature of undergoing public peer review and discussion. This is also reflected on the web pages of ACP and other interactive open access journals of the EGU. The ACP concept is now well established among EGU journals and over the years many newly established journals have followed this example (and future journals will continue to do so, Ervens et al., 2023).

In an e-mail of 18 September 2000, A. Richter wrote that a "meeting of the 'younger and wilder' atmospheric scientists under the lead of Ulrich Pöschl and Paul Crutzen regarding the launch of a new EGS journal on atmospheric chemistry took place on 15 September in Mainz"; this meeting was the birthplace of the new journal ACP (Dingwell et al., 2011). The journal ACP was founded in 2000 with Paul as a member of the advisory board. At that time, the development and success of ACP could hardly have been foreseen.

Today, ACP is a very well established and highly ranked scientific journal. Starting with only 7 published papers in 2001 (34 in 2002 and 158 in 2003) the number of papers published by ACP increased steadily until 2010, when more than 800 papers were published. In recent years, regularly more than 800 papers per year appear in ACP (Pöschl, 2012; Ervens et al., 2023). In Paul's words: "It has been an amazing journey: over a short period of merely a decade, a novel idea originating from Uli Pöschl and developed by an enthusiastic group of hundreds of scientists, created a new way of scientific publishing and communication, initially covering the fields of atmospheric chemistry and physics. The example has since been followed by many successors in other disciplines, with more to come." (Dingwell et al., 2011).

## 4 Anthropocene

With the Anthropocene concept (Crutzen, 2002; Crutzen and Steffen, 2003; Crutzen and Müller, 2019; Benner et al., 2021), Paul expressed his insight that humanity is indeed changing the planet as a whole and should take responsibility for its development. He actively advocated this concept until recently (Fig. 4). The Anthropocene concept also led to the development of the iconic "great acceleration" figures (see e.g., Fig. 2 in Fishman et al., 2023) that show increases in population, in greenhouse gases, in fertiliser consumption, and many other signatures of human impact on the Earth system since the industrial revolution (Steffen et al., 2007).

In 2009, the Anthropocene Working Group (AWG, Fig. 4) was established within the Subcommission on Quaternary Stratigraphy as an interdisciplinary research group dedicated to formalising the Anthropocene as the current geologic time epoch and, more generally, to studying the Anthropocene as a geological time unit. The term Anthropocene became popular after Paul Crutzen proposed it spontaneously at a conference in 2000 to refer to the current epoch (Benner et al., 2021; Müller, 2022; Fishman et al., 2023). Today, the evaluation of the Anthropocene as a formal unit in the geological timescale continues (Ellis et al., 2016; Zalasiewicz et al., 2017; Luciano, 2022; Fishman et al., 2023).

## 5 The impact of Paul Crutzen on atmospheric chemistry and physics

Paul Crutzen had very broad scientific interests and an enormous influence on science; his interests covered practically the entire atmosphere from the ground to the mesosphere. The body of Paul's scientific work is too extensive and broad to be covered here, but some examples have been given above. At the same time, the appreciation of Paul's personality would be incomplete without acknowledging his interest and care for the private lives of the people around him, especially his family.

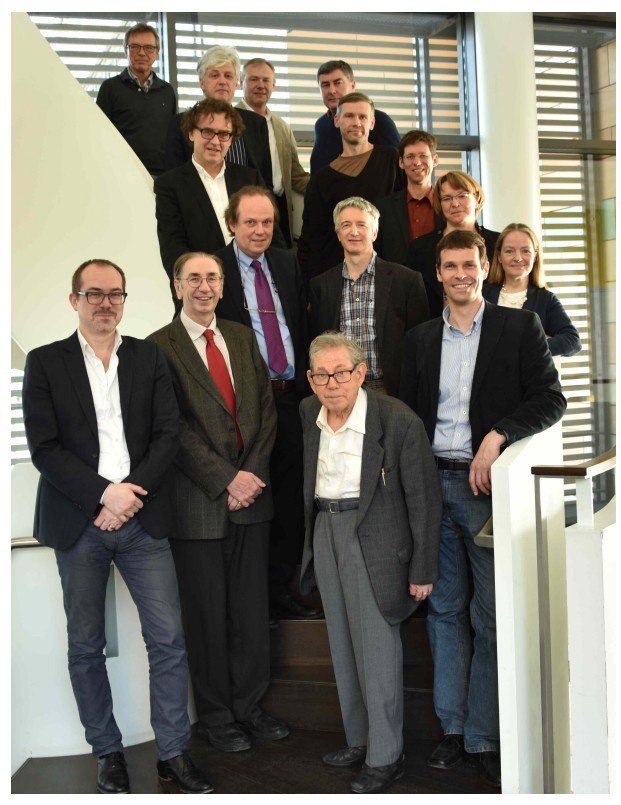

**Figure 4.** Meeting of the Anthropocene Working Group (AWG) at the Max Planck Institute for Chemistry in March 2017. The people on the picture, clockwise from bottom left to bottom right: Franz Mauelshagen, Institut für transformative Nachhaltigkeitsforschung (IASS) Potsdam; Colin Waters, University of Leicester and AWG; Jürgen Renn, MPI für Wissenschaftsgeschichte, Berlin; Bernd Scherer, Haus der Kulturen der Welt (HKW), Berlin; Jos Lelieveld, MPI für Chemie, Mainz; Reinhold Leinfelder, Freie Universität Berlin and AWG; Davor Vidas, Fridtjof Nansen Institut, Oslo and AWG; Mark Williams, University of Leicester and AWG; Christoph Rosol, HKW und MPI für Wissenschaftsgeschichte, Berlin; Mark Lawrence, IASS, Potsdam; Susanne Benner, MPI für Chemie, Mainz; Jan Zalasiewicz, University of Leicester and AWG; Astrid Kaltenbach, MPI für Chemie, Mainz; Uli Pöschl, MPI für Chemie, Mainz; and Paul J. Crutzen, MPI für Chemie, Mainz and AWG. (Picture by S. Schweller, MPI for Chemistry).

On top of his contribution to science, he was also a key figure in the development of an entirely new approach to scientific publishing that began with the journal Atmospheric Chemistry and Physics (ACP). When ACP was founded in 2000, it was unique in featuring public discussion of published preprints and, furthermore, open access to finally accepted and published papers. The 20st anniversary of this journal is celebrated in this special issue. Paul's legacy is honoured in ACP in form of the ACP Paul Crutzen publication award, which was created to recognise an outstanding publication in ACP in a particular year. The first prize was awarded in 2021.

Paul's work not only had a profound impact on the scientific world, but also influenced the environmental politics of many countries. His scientific work will continue to provide guidance for the evolution of science. Likewise, his ideas will continue to have a strong influence on future global policies needed to halt the warming of Earth's climate and the destruction of our planet as we know it.

*Data availability.* not applicable

*Author contributions.* U.P., T.K., T.P., K.C., and R.M. all contributed to putting together the material for this paper and to writing the manuscript.

*Competing interests.* K.C. and R.M. are editors of ACP; T.K. and U.P. are members of ACP's advisory board. Otherwise, the authors declare no competing interest.

*Acknowledgements.* First of all, we thank Terttu Crutzen for reading and commenting on this manuscript. Further, we thank Susanne Benner, John W. Birks, Guy Brasseur, Sylvia Crutzen, Barbara Ervens, Astrid Kaltenbach, A. R. Ravishankara, and Alan Robock for comments on this paper. We thank Paul's daughter Ilona and his grandson Jamie Paul for providing pictures from the family archives. Figure 2 is courtesy of Jens-Uwe Grooß.

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
