# Peer review of "Paul J. Crutzen – a pioneer in Earth system science and a founding member of the journal "Atmospheric Chemistry and Physics""

_EGUsphere, 2023_

## Referee Comment (RC1)

**Review of Müller et al. (2023), Paul J. Crutzen – a pioneer in Earth system science and a founding member of the journal "Atmospheric Chemistry and Physics"**

This is a great paper and certainly should be published.  But Paul's role in nuclear winter is incorrect, and he only contributed to one proposed climate intervention scheme, injecting $SO_2$ into the stratosphere to create an aerosol cloud, not to geoengineering in general.  And we now use climate intervention and not "geoengineering" to refer to this topic, and Paul's paper used "climate intervention" in the title.

Also, all the praise for *ACP* needs to be tempered with a big problem it created, that of Discussions papers that are not peer-reviewed, but have DOIs and end up being referenced. Please see below for details and also respond to the 8 comments in the attached manuscript.

It is not correct that Paul Crutzen "pioneered the concept now known as 'nuclear winter.'"  But his work on smoke from fires after nuclear war (Crutzen and Birks, 1982) did lead directly to the theory of nuclear winter calculated by other scientists (Turco et al., 1983; Aleksandrov and Stenchikov, 1983; Robock, 1984; Covey et al., 1984).  While Crutzen and Birks (1982) pointed out that there would be fires after nuclear war which would produce so much smoke that it would be dim at Earth's surface, hence the title "Twilight at Noon," that paper never mentioned how surface temperature would change.  As John Birks told me last year, they studied tropospheric smoke, not stratospheric, and they knew that the reduction of sunlight would be balanced by heating by absorption of sunlight by the black smoke.  It took Turco et al. (1983) and Aleksandrov and Stenchikov (1983), soon followed by Robock (1984) and Covey et al. (1984) to calculate the surface temperature and explain it as nuclear winter.  But of course, they were inspired by Crutzen and Birks.

Crutzen and Birks (1982) write about a lifetime for the smoke of weeks and maybe months, and that the smoke would be removed by rain and dry deposition ("Alternatively these reactions begin to occur after an appreciable fraction of the aerosol loading of the atmosphere has diminished because of removal of the particulate matter by rain or dry deposition.").  In fact, they say, "Our model does not predict significant stratospheric ozone depletion" for the scenario they studied, but they did write, "Finally, we may point out that there is a possibility that even a nuclear war according to Scenario I, in which most $NO_x$ is deposited in the troposphere, may cause ozone depletions in the stratosphere, if the hot fires in the oil and gas production regions become so powerful that the fire plumes penetrate into the stratosphere.  Another means of upward transport may occur when the heavy, dark aerosol layer, initially located in the troposphere, is heated by solar radiation and starts to set up convection and wind systems which will transport an appreciable fraction of the fire effluents into the stratosphere. These speculative thoughts may be pursued further with currently available general circulation models of the atmosphere."

Crutzen and Birks (1982) make no conclusions about the impacts on climate on nuclear war, and did not think it likely.  They only speculate about it and say the answers must await further work. They write, "It may be possible to test the impact of nuclear war on climate with this [referring to Jim Hansen's recent work] and similar models when these are supplied with reasonable estimates of the trace gas and aerosol composition of the earth's atmosphere. Whether the

induced perturbation in the climate system could lead to longer lasting climatic changes will, however, be difficult to predict. In fact, it may seem unlikely that it will take place. The Krakatoa volcanic eruption of 1883 injected quantities of aerosol into the atmosphere comparable to those which would be caused by a nuclear war, and global mean temperatures were affected for only a few years (1). Still, we must be cautious with a prediction as the physical characteristics of the aerosol produced by volcanos and fires are different, and much is still unknown about the fundamentals of climatic changes. For instance, we may ask questions such as whether the earth's albedo would be substantially altered after a nuclear war and thus affect the radiation balance or whether the deposition of soot aerosol on arctic snow and ice and on the glaciers of the Northern Hemisphere might not lead to such heavy snow and ice melting as to cause an irreversible change in one or more important climatic parameters."

Paul did reopen the debate about geoengineering (now known as "climate intervention"), but only one particular kind. He only discussed solar radiation modification, and not carbon dioxide removal. In fact, he only discussed one proposed scheme, "albedo enhancement by stratospheric sulfur injections."

Lines 149-150: You should also mention the problem that open access to ACP preprints creates. They are assigned a DOI, and I have seen many submitted and even published papers that reference Discussions papers. Sometimes these are rejected papers that should never be referenced, and all the others are not peer-reviewed and have issues that were resolved in the final versions. Just saying how wonderful ACP is while ignoring this problem is disingenuous.

I reproduce below two photos of the instructors at "Governing Climate Engineering – A Transdisciplinary Summer School," Max-Planck-Institute for Comparative Public Law and International Law, Heidelberg, Germany, July 12-16, 2010. Unfortunately, there were no women, and you might want to comment on that. Nowadays many women are working on this topic. I can identify the Americans there, David Keith, Phil Rasch, and Alan Robock, and also Tom Peter and, of course, Paul. I am sure you can identify the others, and you might want to consider using one of these photos in the paper. They illustrate Paul's continuous interest in this important research area that he stimulated. They are my photos and you are free to use them. If you want high-resolution versions, please let me know.

**References**

[revised manuscript text omitted]

---

## Referee Comment (RC3)

**Review of Müller et al. (2023), Paul J. Crutzen – a pioneer in Earth system science and a founding member of the journal "Atmospheric Chemistry and Physics"**

This is a great paper and certainly should be published. But Paul's role in nuclear winter is incorrect, and he only contributed to one proposed climate intervention scheme, injecting $SO_2$ into the stratosphere to create an aerosol cloud, not to geoengineering in general. And we now use climate intervention and not "geoengineering" to refer to this topic, and Paul's paper used "climate intervention" in the title.

Also, all the praise for *ACP* needs to be tempered with a big problem it created, that of Discussions papers that are not peer-reviewed, but have DOIs and end up being referenced. Please see below for details and also respond to the 8 comments in the attached manuscript.

It is not correct that Paul Crutzen "pioneered the concept now known as 'nuclear winter.'" But his work on smoke from fires after nuclear war (Crutzen and Birks, 1982) did lead directly to the theory of nuclear winter calculated by other scientists (Turco et al., 1983; Aleksandrov and Stenchikov, 1983; Robock, 1984; Covey et al., 1984). While Crutzen and Birks (1982) pointed out that there would be fires after nuclear war which would produce so much smoke that it would be dim at Earth's surface, hence the title "Twilight at Noon," that paper never mentioned how surface temperature would change. As John Birks told me last year, they studied tropospheric smoke, not stratospheric, and they knew that the reduction of sunlight would be balanced by heating by absorption of sunlight by the black smoke. It took Turco et al. (1983) and Aleksandrov and Stenchikov (1983), soon followed by Robock (1984) and Covey et al. (1984) to calculate the surface temperature and explain it as nuclear winter. But of course, they were inspired by Crutzen and Birks.

Crutzen and Birks (1982) write about a lifetime for the smoke of weeks and maybe months, and that the smoke would be removed by rain and dry deposition ("Alternatively these reactions begin to occur after an appreciable fraction of the aerosol loading of the atmosphere has diminished because of removal of the particulate matter by rain or dry deposition."). In fact, they say, "Our model does not predict significant stratospheric ozone depletion" for the scenario they studied, but they did write, "Finally, we may point out that there is a possibility that even a nuclear war according to Scenario I, in which most $NO_x$ is deposited in the troposphere, may cause ozone depletions in the stratosphere, if the hot fires in the oil and gas production regions become so powerful that the fire plumes penetrate into the stratosphere. Another means of upward transport may occur when the heavy, dark aerosol layer, initially located in the troposphere, is heated by solar radiation and starts to set up convection and wind systems which will transport an appreciable fraction of the fire effluents into the stratosphere. These speculative thoughts may be pursued further with currently available general circulation models of the atmosphere."

Crutzen and Birks (1982) make no conclusions about the impacts on climate on nuclear war, and did not think it likely. They only speculate about it and say the answers must await further work. They write, "It may be possible to test the impact of nuclear war on climate with this [referring to Jim Hansen's recent work] and similar models when these are supplied with reasonable estimates of the trace gas and aerosol composition of the earth's atmosphere. Whether the

induced perturbation in the climate system could lead to longer lasting climatic changes will, however, be difficult to predict. In fact, it may seem unlikely that it will take place. The Krakatoa volcanic eruption of 1883 injected quantities of aerosol into the atmosphere comparable to those which would be caused by a nuclear war, and global mean temperatures were affected for only a few years (1). Still, we must be cautious with a prediction as the physical characteristics of the aerosol produced by volcanos and fires are different, and much is still unknown about the fundamentals of climatic changes. For instance, we may ask questions such as whether the earth's albedo would be substantially altered after a nuclear war and thus affect the radiation balance or whether the deposition of soot aerosol on arctic snow and ice and on the glaciers of the Northern Hemisphere might not lead to such heavy snow and ice melting as to cause an irreversible change in one or more important climatic parameters."

Paul did reopen the debate about geoengineering (now known as "climate intervention"), but only one particular kind. He only discussed solar radiation modification, and not carbon dioxide removal. In fact, he only discussed one proposed scheme, "albedo enhancement by stratospheric sulfur injections."

Lines 149-150: You should also mention the problem that open access to ACP preprints creates. They are assigned a DOI, and I have seen many submitted and even published papers that reference Discussions papers. Sometimes these are rejected papers that should never be referenced, and all the others are not peer-reviewed and have issues that were resolved in the final versions. Just saying how wonderful ACP is while ignoring this problem is disingenuous.

I reproduce below two photos of the instructors at "Governing Climate Engineering – A Transdisciplinary Summer School," Max-Planck-Institute for Comparative Public Law and International Law, Heidelberg, Germany, July 12-16, 2010. Unfortunately, there were no women, and you might want to comment on that. Nowadays many women are working on this topic. I can identify the Americans there, David Keith, Phil Rasch, and Alan Robock, and also Tom Peter and, of course, Paul. I am sure you can identify the others, and you might want to consider using one of these photos in the paper. They illustrate Paul's continuous interest in this important research area that he stimulated. They are my photos and you are free to use them. If you want high-resolution versions, please let me know.

**Correspondence:** Rolf Müller <ro.mueller@fz.juelich.de>

first versus one of the first. e.g., Hal Johnston and NOx catalyzed strat ozone loss.

[revised manuscript text omitted]

this had a major impact since NOx emissions from SSTs was the first strat ozone depletion issue that was studies.

$$NO_2 + O \quad \rightarrow \quad NO + O_2 \qquad \text{(2)}$$

where the sum of NO and $NO_2$ is defined as $NO_x = NO + NO_2$ and O indicates an oxygen atom in ground-state ($O(^3P)$).

The discovery of this mechanism in 1970 was a crucial step towards facilitating a quantitative description of the chemistry of the stratospheric ozone layer; prior to Paul's discovery, the dominant catalytic loss cycle of stratospheric ozone ($D_N$, through reactions 1 and 2) was not known (Fig. 2). It is now known that $D_O$: the reaction $O_3 + O \rightarrow 2O_2$, originally proposed by Chapman (1930), is only a minor sink of stratospheric ozone and that $HO_x$-induced catalysis only is important close to the tropopause and above about 45 km (e.g., Portmann et al., 2012; Müller, 2021, see also Fig. 2). The recognition that chlorine ($D_{Cl}$) also catalytically contributes to stratospheric ozone loss came a few years after the discovery of the $NO_x$-induced cycle (Molina and Rowland, 1974).

Paul was engaged in investigations on the impact of a possible fleet of supersonic planes (and the emissions of $NO_x$ caused by this fleet) on stratospheric ozone (Johnston, 1971; Crutzen, 1972). He also investigated how active nitrogen compounds can enter the stratosphere (Crutzen and Ehhalt, 1977; Müller, 2021). He initiated the first studies on the budget of $N_2O$ in the atmosphere and how it is influenced by human activity (Crutzen and Ehhalt, 1977).

Paul made fundamental discoveries in tropospheric chemistry (e.g., Fishman and Crutzen, 1978; Fishman et al., 1979b; Crutzen and Zimmermann, 1991; Fishman et al., 2023) showing that tropospheric chemical processes provide a much larger source of tropospheric ozone than downward transport of ozone from the stratosphere. This work led him to realise the im-

This was also done by Chameides and Walker in 1973. The work of Crutzen and C&W were the key studies that led to the realization that tropospheric ozone is mostly produced in the troposphere.

[revised manuscript text omitted]

---

## Author Comment (AC1)

**Reply to Alan Robock**

We thank Alan Robock very much for his review – this is very helpful. His comments in the reply are in *italics* and the reply by the authors are in roman font. We reply to the comments on Paul Crutzen's work and on ACP in detail below and also explain the changes made to the manuscript in response.

*This is a great paper and certainly should be published. But Paul's role in nuclear winter is incorrect, and he only contributed to one proposed climate intervention scheme, injecting $SO_2$ into the stratosphere to create an aerosol cloud, not to geoengineering in general. And we now use climate intervention and not "geoengineering" to refer to this topic, and Paul's paper used "climate intervention" in the title.*

We agree that this article should accurately portray Paul's role in establishing the concepts of "nuclear winter" and "climate intervention". In response we have altered the manuscript, added some discussion and included more references.

A related change to our manuscript, not directly suggested in the comment is that we now mention in the paper that the "Future of Life Institute" has presented John Birks, Paul Crutzen, Jeannie Peterson, Alan Robock, Carl Sagan, Georgiy Stenchikov, Brian Toon and Richard Turco with the 2022 "Future of Life Award" for reducing the risk of nuclear war by developing and popularising the science of nuclear winter.

*Also, all the praise for ACP needs to be tempered with a big problem it created, that of Discussions papers that are not peer-reviewed, but have DOIs and end up being referenced. Please see below for details and also respond to the 8 comments in the attached manuscript.*

This comment is discussed in detail below.

**"Nuclear winter" and "geoengineering"**

*It is not correct that Paul Crutzen "pioneered the concept now known as 'nuclear winter'." But his work on smoke from fires after nuclear war (Crutzen and Birks, 1982) did lead directly to the theory of nuclear winter calculated by other scientists (Turco et al., 1983; Aleksandrov and Stenchikov, 1983; Robock, 1984; Covey et al., 1984). While Crutzen and Birks (1982) pointed out that there would be fires after nuclear war which would produce so much smoke that it would be dim at Earth's surface, hence the title "Twilight at Noon", that paper never mentioned how surface temperature would change. As John Birks told me last year, they studied tropospheric smoke, not stratospheric, and they knew that the reduction of sunlight would be balanced by heating by absorption of sunlight by the black smoke. It took Turco et al. (1983) and Aleksandrov and Stenchikov (1983), soon followed by Robock (1984) and Covey et al. (1984) to calculate the surface temperature and explain it as nuclear winter. But of course, they were inspired by Crutzen and Birks. Crutzen and Birks (1982) write about a lifetime for the smoke of weeks and maybe months, and that the smoke would be removed by rain and dry deposition ("Alternatively these reactions begin to occur after*

*an appreciable fraction of the aerosol loading of the atmosphere has diminished because of removal of the particulate matter by rain or dry deposition.").  In fact, they say, "Our model does not predict significant stratospheric ozone depletion" for the scenario they studied, but they did write, "Finally, we may point out that there is a possibility that even a nuclear war according to Scenario I, in which most NOx is deposited in the troposphere, may cause ozone depletions in the stratosphere, if the hot fires in the oil and gas production regions become so powerful that the fire plumes penetrate into the stratosphere.  Another means of upward transport may occur when the heavy, dark aerosol layer, initially located in the troposphere, is heated by solar radiation and starts to set up convection and wind systems which will transport an appreciable fraction of the fire effluents into the stratosphere.  These speculative thoughts may be pursued further with currently available general circulation models of the atmosphere."  Crutzen and Birks (1982) make no conclusions about the impacts on climate on nuclear war, and did not think it likely.  They only speculate about it and say the answers must await further work.  They write, "It may be possible to test the impact of nuclear war on climate with this [referring to Jim Hansen's recent work] and similar models when these are supplied with reasonable estimates of the trace gas and aerosol composition of the earth's atmosphere.  Whether the induced perturbation in the climate system could lead to longer lasting climatic changes will, however, be difficult to predict.  In fact, it may seem unlikely that it will take place.  The Krakatoa volcanic eruption of 1883 injected quantities of aerosol into the atmosphere comparable to those which would be caused by a nuclear war, and global mean temperatures were affected for only a few years (1).  Still, we must be cautious with a prediction as the physical characteristics of the aerosol produced by volcanos and fires are different, and much is still unknown about the fundamentals of climatic changes.  For instance, we may ask questions such as whether the earth's albedo would be substantially altered after a nuclear war and thus affect the radiation balance or whether the deposition of soot aerosol on arctic snow and ice and on the glaciers of the Northern Hemisphere might not lead to such heavy snow and ice melting as to cause an irreversible change in one or more important climatic parameters".*

We agree with these points and now say in the paper:

Abstract: "His work on smoke from fires a after nuclear war inspired new research on a concept now known as "nuclear winter"."

Body of the paper:

"Moreover, he was involved in the first studies on the global effects of a thick smoke layer in the atmosphere produced by fires caused by a possible nuclear war ..."

and

"Paul's work on smoke from fires after a possible nuclear war and the absorption of sunlight by the smoke (Crutzen and Birks, 1982; Birks and Crutzen, 1983) introduced the concept that the use of nuclear weapons would have global impacts that go much beyond the more obvious direct effects.  This work inspired substantial research activity – Turco et al. (1983) and Aleksandrov and Stenchikov (1983), soon followed by Robock (1984) and Covey et al. (1984)

calculated the surface temperature response to fires after a nuclear war and introduced the term "nuclear winter (see also Robock et al., 2023)"."

*Paul did reopen the debate about geoengineering (now known as "climate intervention"), but only one particular kind. He only discussed solar radiation modification, and not carbon dioxide removal. In fact, he only discussed one proposed scheme, "albedo enhancement by stratospheric sulfur injections".*

We agree and have reformulated the respective papers of the paper.

Abstract:

"He also initiated the reopening of the debate on "geoengineering" – a concept now referred to as "climate intervention". "

We now say in the paper:

"...Paul has also sparked a new debate by breaking the taboo regarding a possible cooling of the climate by increasing the earth's albedo through stratospheric sulphur injections (Crutzen, 2006)."

Paul also initiated a new discussion on the question of "geoengineering" (now known as "climate intervention") by discussing the role of albedo enhancements caused by stratospheric sulphur injections (Crutzen, 2006)."

and

"These two papers are on the global consequences of a possible nuclear exchange (nuclear winter, Robock et al., 2023) and on climate intervention (Visioni et al., 2023)."

**"Discussion Papers"**

*Lines 149-150: You should also mention the problem that open access to ACP preprints creates. They are assigned a DOI, and I have seen many submitted and even published papers that reference Discussions papers. Sometimes these are rejected papers that should never be referenced, and all the others are not peer-reviewed and have issues that were resolved in the final versions. Just saying how wonderful ACP is while ignoring this problem is disingenuous.*

We agree with Alan Robock that peer reviewed published papers should not be confused with unrefereed preprints. However, this confusion should have been prevented by the two names "Atmospheric Chemistry and Physics" (ACP, for peer reviewed journal articles) and "Atmospheric Chemistry and Physics Discussions" (ACPD, for preprints under discussion). In any case, these two types of publications are different. We also admit, that when the concept of public discussion of submitted manuscripts was introduced, there was initially some confusion in the community about the status of discussion papers (likely because the idea was new at that time).

However, we think that referring to preprints under consideration at ACP is very similar to referring to manuscripts/preprints on arXiv.org and other preprint servers. The only difference being that ACPD papers were under review for the journal, while preprints on arXiv may or may not be under review in a journal. In particular the status of submitted papers/preprints has been further clarified since early 2023 by retiring ACPD and introducing the preprint server EGUsphere for submitted papers for journals like ACP. EGUsphere is

a repository for both standalone preprints (much like arXiv.org) and preprints that are under review in ACP. Retiring ACPD was a step taken to partly address concerns like the one expressed in this comment. In particular, the EGUsphere doi makes the preprint status clearer, while the doi also ensures that people's free speech and intellectual ideas are protected and properly archived – a paper on EGUsphere remains a preprint.

Further, the sytem of manuscript reviewing and editorial decisions at ACP was evaluated independently (Bornmann et al., 2010, 2011). There will also be a paper in this special issue (Ervens et al., 2023) where the publishing process in ACP is discussed in much more detail than appropriate and possible here.

Nonetheless, in order to mention the problem that open access to ACP preprints possibly creates, we have introduced the following text in the paper: "Initially, when the concept of public public peer-review and a public discussion of submitted manuscripts was introduced, there was some confusion in the community about the status of discussion papers. Since early 2023, to clarify the status of submitted papers (which are not peer-reviewed), the term 'Atmospheric Chemistry and Physics Discussions' (ACPD) is no longer used and papers submitted to ACP appear on the preprint server EGUsphere (after an initial screening)."

**References**

Aleksandrov, V. V. and Stenchikov, G. L.: On the modeling of the climatic consequences of the nuclear war, in: Proc. Applied Math., p. 21, Computing Centre, USSR Academy of Sciences, Moscow, 1983.

Birks, J. W. and Crutzen, P. J.: Atmospheric effects of a nuclear war, Chemistry in Britain, pp. 927–930, 1983.

Bornmann, L., Marx, W., Schier, H., Thor, A., and Daniel, H.-D.: From black box to white box at open access journals: predictive validity of manuscript reviewing and editorial decisions at *Atmospheric Chemistry and Physics*, Research Evaluation, 19, 105–118, https://doi.org/10.3152/095820210X510089, 2010.

Bornmann, L., Schier, H., Marx, W., and Daniel, H.-D.: Is Interactive Open Access Publishing Able to Identify High-Impact Submissions? A Study on the Predictive Validity of *Atmospheric Chemistry and Physics* by Using Percentile Rank Classes, Journal of the American Society for Information Science and Technology, pp. 62–71, https://doi.org/10.1002/asi, 2011.

Covey, C., Thompson, S., and Schneider, S. H.: Global atmospheric effects of massive smoke injections from a nuclear war: Results from general circulation model simulations, Nature, 308, 21–25, 1984.

Crutzen, P. J.: Albedo enhancements by stratospheric sulfur injections: a contribution to resolve a policy dilemma? An Editorial Essay, Clim. Change, 77, 211–219, 2006.

Crutzen, P. J. and Birks, J. W.: The atmosphere after a nuclear war: Twilight at Noon., Ambio, 2&3, 114–125, 1982.

Ervens, B., Pöschl, U., and Carslaw, K.: Interactive open access publishing with multi-stage open peer review in the European Geosciences Union (EGU): more than two decades of successful transparent, self-regulating, and community-based scientific exchange and quality assurance, Atmos. Chem. Phys., in preparation, 2023.

Robock, A.: Snow and ice feedbacks prolong effects of nuclear winter, Nature, 310, 667–670, 1984.

Robock, A., Xia, L., Harrison, C. S., Coupe, J., Toon, O. B., and Bardeen, C. G.: Opinion: How fear of nuclear winter has helped save the world, so far, Atmos. Chem. Phys., 23, 6691–6701, https://doi.org/10.5194/acp-23-6691-2023, 2023.

Turco, R. P., Toon, O. B., Ackerman, T. P., Pollack, J. B., and Sagan, C.: Nuclear winter: Global consequences of multiple nuclear explosions, Science, 222, 1283–1292, https://doi.org/10.1126/science.222.4630.1283, 1983.

Visioni, D., Kravitz, B., Robock, A., Tilmes, S., Haywood, J., Boucher, O., Lawrence, M., Irvine, P., Niemeier, U., Xia, L., Chiodo, G., Lennard, C., Watanabe, S., Moore, J. C., and Muri, H.: Opinion: The scientific and community-building roles of the Geoengineering Model Intercomparison Project (GeoMIP) – past, present, and future, Atmos. Chem. Phys., 23, 5149–5176, https://doi.org/10.5194/acp-23-5149-2023, 2023.

---

## Author Comment (AC3)

**Reply to second review**

**General**

Many of the comments in the review are by Alan Robock and these comments are addressed in the reply to Alan Robock. There are however some new comments in the review (red comments) which are repeated here (in red). We are very grateful for these comments. The reply and the associated changes in the text of the manuscript are given below (in black).

**Reply in detail**

**Comment 1**

(abstract) First versus one of the first. e.g., Hal Johnston and NOx catalyzed strat ozone loss.

We have emphasised the contribution by Hal Johnston now in the following text in the paper:

"The recognition of the strong effect of $NO_x$ on stratospheric ozone had a major impact, since emissions of $NO_x$ caused by a possible fleet of supersonic planes was the first stratospheric ozone depletion issue that was studied (Johnston, 1971; Crutzen, 1972)."

We know that Paul Crutzen (in conversations) made the point that his 1970 contribution should not be forgotten (Crutzen, 1970, although he felt that sometimes it is), so we decided to not change the wording here.

**Comment 2**

(geological institutions): what does this mean?

We agree and have removed the comment on geological institutions.

**Comment 3**

(football) for American audience- Soccer.

We agree and have added a footnote stating "Football is the game referred to in American English as soccer".

**Comment 4**

(tropospheric ozone): This was also done by Chameides and Walker in 1973. The work of Crutzen and C&W were the key studies that led to the realization that tropospheric ozone is mostly produced in the troposphere.

We agree that the work of Chameides and Walker (1973) needs to be mentioned when talking about tropospheric chemistry. In response to the comment we have now expanded and clarified the discussion; the relevant text reads now in the manuscript:

"Tropospheric chemistry, and in particular the chemical production of of ozone in the troposphere were topics of great importance to Paul (Crutzen, 1996; Fishman et al., 2023). The radical OH (Levy, 1971; Crutzen and Zimmermann, 1991; Crutzen, 1996) is responsible for the oxidation of $CH_4$ (and many other compounds emitted into the atmosphere). It was found that in environments containing sufficient NO, the methane oxidation chain could produce ozone in large quantities – large compared to the downward flux of ozone from the stratosphere to the troposphere – (Crutzen, 1973; Chameides and Walker, 1973; Crutzen, 1974). Paul's work and the work of Chameides and Walker (1973) were the key studies that led to the realisation that tropospheric ozone is mostly chemically produced in the troposphere. In the early seventies, very little was known about homogeneous and heterogeneous reactions affecting the methane oxidation chain in the troposphere, so that initial conclusions remained uncertain (Crutzen, 1974). However some years later, together with Jack Fishman and Susan Solomon, Paul presented observational evidence for a strong in situ tropospheric ozone production (Fishman and Crutzen, 1978; Fishman et al., 1979). Later, Paul's work on tropospheric ozone led him to realise the importance of biomass-burning for the chemistry of the atmosphere (Crutzen et al., 1979; Crutzen and Andreae, 1990)."

**Comment 5**

(NOx cycle): this had a major impact since NOx emissions from SSTs was the first strat ozone depletion issue that was studied.

We agree and have extended the text on the SST issue. It reads now:

"The recognition of the strong effect of $NO_x$ on stratospheric ozone had a major impact, since emissions of $NO_x$ caused by a possible fleet of supersonic planes was the first stratospheric ozone depletion issue that was studied (Johnston, 1971; Crutzen, 1972)."

**Comment 6**

(Open review and open access): You may want to qualify this. Faraday Transactions and the Proceedings of the Combustion Institute did this long time ago, but in a Q&A format.

We agree with this comment and in response we have added the following text to the manuscript: 'Scientific discussions have been documented before the invention of ACP; for example in the "Electronic Transactions in Artificial Intelligence" (ETAI) and the "Journal of Interactive Media in Education" (JIME) (Pöschl, 2012, section "Comparison to earlier initiatives with two- or multi-stage open peer review"). Further, the review process in "Faraday Transactions" and in the "Proceedings of the Combustion Institute" is "open", in the sense that these journals have a long history of meetings and their subsequent publication of the discussion. Faraday Discussions collects questions and answers through delegate discussion during meetings (rather than online or through a text forum); a discussion which is then edited and published alongside the articles in each volume. A similar procedure is followed by the Combustion Institute (see Nicovich and Ravishankara, 1982, for an example). The interactive open access process (as we call it today) with a multi-stage public peer review as practised in ACP, however, had not been introduced in scientific publishing prior to the launch of ACP (Pöschl, 2012).'

**References**

Chameides, W. and Walker, J.: A photochemical theory of tropospheric ozone, J. Geophys. Res., 78, 8751–8760, URL https://doi.org/10.1029/JC078i036p08751, 1973.

Crutzen, P. J.: The influence of nitrogen oxides on the atmospheric ozone content, Q. J. R. Meteorol. Soc., 96, 320–325, 1970.

Crutzen, P. J.: SSTs–a threat to the earth's ozone shield, Ambio, 1, 41–51, 1972.

Crutzen, P. J.: A discussion of the chemistry of some minor constituents in the stratosphere and troposphere, Pure Appl. Geophys., 106-108, 1385–1399, 1973.

Crutzen, P. J.: Photochemical reactions initiated by and influencing ozone in unpolluted tropospheric air, Tellus, 26, 48–57, 1974.

Crutzen, P. J.: My Life with $O_3$, $NO_x$ and other $YZO_x$ compounds, Angew. Chem. Intern. Ed., 35, 1747–1871, 1996.

Crutzen, P. J. and Andreae, M. O.: Biomass burning in the tropics – Impact on atmospheric chemistry and biogeochemical cycles, Science, 250, 1669–1678, 1990.

Crutzen, P. J. and Zimmermann, P. H.: The changing photochemistry of the troposphere, Tellus, 43, 136–151, https://doi.org/10.3402/tellusb.v43i4.15403, 1991.

Crutzen, P. J., Heidt, L. E., Krasnec, J. P., Pollock, W. H., and Seiler, W.: Biomass burning as a source of atmospheric gases CO, $H_2$, $N_2O$, NO, $CH_3Cl$ and COS, Nature, 282, 253–256, 1979.

Fishman, J. and Crutzen, P.: The origin of ozone in the troposphere, Nature, 274, 855–858, URL https://doi.org/10.1038/274855a0, 1978.

Fishman, J., Solomon, S., and Crutzen, P. J.: Observational and theoretical evidence in support of a significant in situ photochemical source of tropospheric ozone, Tellus, 31, 432–446, 1979.

Fishman, J., Birks, J. W., Graedel, T. E., Steffen, W., Burrows, J. P., Howard, C. J., and Wayne, R. P.: A Tribute to Paul Crutzen (1933–2021): The Pioneering Atmospheric Chemist Who Provided New Insight into the Concept of Climate Change, Bull. Am. Meteorol. Soc., 104, E77 – E95, https://doi.org/10.1175/BAMS-D-21-0311.1, 2023.

Johnston, H.: Reduction of stratospheric ozone by nitrogen oxide catalysts from supersonic transport exhaust, Science, 173, 517–522, 1971.

Levy, H.: Normal atmosphere: Large radical and formaldehyde concentrations predicted, Science, 173, 141–143, https://doi.org/10.1126/science.173.3992.141, 1971.

Nicovich, J. M. and Ravishankara, A. R.: A study of the reaction of $O(^3P)$ with ethylene, Nineteenth Symposium (International) on Combustion/The Combustion Institute, pp. 23–30, 1982.

Pöschl, U.: Multi-Stage Open Peer Review: Scientific Evaluation Integrating the Strengths of Traditional Peer Review with the Virtues of Transparency and Self-Regulation, Frontiers in Computational Neuroscience, 6, https://doi.org/10.3389/fncom.2012.00033, 2012.

---

## Author Response (AR2)

**Reply to reviewers comments**

We thank Guy Brasseur, A. R. Ravishankara and Alan Robock very much for their valuable comments that helped improving the paper. All the comments are repeated below and the corresponding changes to the manuscript are reported also. The text in this reply is practically identical to the text in the response published in the discussion of the paper (but see also page 4 of this reply).

Please also note that a few references were added (Carslaw et al., 1993; Tabazadeh et al., 1994; Solomon, 1999; Kirner et al., 2015); see also page 6 of the revised manuscript.

**Reply to Alan Robock**

We thank Alan Robock very much for his review – this is very helpful. His comments in the reply are in *italics* and the reply by the authors are in roman font. We reply to the comments on Paul Crutzen's work and on ACP in detail below and also explain the changes made to the manuscript in response.

Regarding ACP(D) we use a slighlty different wording than in the original response; we say now in the paper "...discussion papers are preprints similar to the manuscripts posted on other preprint servers like arXiv.org but with the additional feature of undergoing public peer review and discussion. This is also reflected on the web pages of ACP and other interactive open access journals of the European Geosciences Union (EGU)."

*This is a great paper and certainly should be published. But Paul's role in nuclear winter is incorrect, and he only contributed to one proposed climate intervention scheme, injecting $SO_2$ into the stratosphere to create an aerosol cloud, not to geoengineering in general. And we now use climate intervention and not "geoengineering" to refer to this topic, and Paul's paper used "climate intervention" in the title.*

We agree that this article should accurately portray Paul's role in establishing the concepts of "nuclear winter" and "climate intervention". In response we have altered the manuscript, added some discussion and included more references.

A related change to our manuscript, not directly suggested in the comment is that we now mention in the paper that the "Future of Life Institute" has presented John Birks, Paul Crutzen, Jeannie Peterson, Alan Robock, Carl Sagan, Georgiy Stenchikov, Brian Toon and Richard Turco with the 2022 "Future of Life Award" for reducing the risk of nuclear war by developing and popularising the science of nuclear winter.

*Also, all the praise for ACP needs to be tempered with a big problem it created, that of Discussions papers that are not peer-reviewed, but have DOIs and end up being referenced. Please see below for details and also respond to the 8 comments in the attached manuscript.*

This comment is discussed in detail below.

**"Nuclear winter" and "geoengineering"**

*It is not correct that Paul Crutzen "pioneered the concept now known as 'nuclear winter'." But his work on smoke from fires after nuclear war (Crutzen and Birks, 1982) did lead directly to the theory of nuclear winter calculated by other scientists (Turco et al., 1983; Aleksandrov and Stenchikov, 1983; Robock, 1984; Covey et al., 1984). While Crutzen and Birks (1982) pointed out that there would be fires after nuclear war which would produce so much smoke that it would be dim at Earth's surface, hence the title "Twilight at Noon", that paper never mentioned how surface temperature would change. As John Birks told me last year, they studied tropospheric smoke, not stratospheric, and they knew that the reduction of sunlight would be balanced by heating by absorption of sunlight by the black smoke. It took Turco et al. (1983) and Aleksandrov and Stenchikov (1983), soon followed by Robock (1984) and Covey et al. (1984) to calculate the surface temperature and explain it as nuclear winter. But of course, they were inspired by Crutzen and Birks. Crutzen and Birks (1982) write about a lifetime for the smoke of weeks and maybe months, and that the smoke would be removed by rain and dry deposition ("Alternatively these reactions begin to occur after an appreciable fraction of the aerosol loading of the atmosphere has diminished because of removal of the particulate matter by rain or dry deposition."). In fact, they say, "Our model does not predict significant stratospheric ozone depletion" for the scenario they studied, but they did write, "Finally, we may point out that there is a possibility that even a nuclear war according to Scenario I, in which most NOx is deposited in the troposphere, may cause ozone depletions in the stratosphere, if the hot fires in the oil and gas production regions become so powerful that the fire plumes penetrate into the stratosphere. Another means of upward transport may occur when the heavy, dark aerosol layer, initially located in the troposphere, is heated by solar radiation and starts to set up convection and wind systems which will transport an appreciable fraction of the fire effluents into the stratosphere. These speculative thoughts may be pursued further with currently available general circulation models of the atmosphere." Crutzen and Birks (1982) make no conclusions about the impacts on climate on nuclear war, and did not think it likely. They only speculate about it and say the answers must await further work. They write, "It may be possible to test the impact of nuclear war on climate with this [referring to Jim Hansen's recent work] and similar models when these are supplied with reasonable estimates of the trace gas and aerosol composition of the earth's atmosphere. Whether the induced perturbation in the climate system could lead to longer lasting climatic changes will, however, be difficult to predict. In fact, it may seem unlikely that it will take place. The Krakatoa volcanic eruption of 1883 injected quantities of aerosol into the atmosphere comparable to those which would be caused by a nuclear war, and global mean temperatures were affected for only a few years (1). Still, we must be cautious with a prediction as the physical characteristics of the aerosol produced by volcanos and fires are different, and much is still unknown about the fundamentals of climatic changes. For instance, we may ask questions such as whether the earth's albedo would be substantially altered after a nuclear war*

*and thus affect the radiation balance or whether the deposition of soot aerosol on arctic snow and ice and on the glaciers of the Northern Hemisphere might not lead to such heavy snow and ice melting as to cause an irreversible change in one or more important climatic parameters".*

We agree with these points and now say in the paper:

Abstract: "His work on smoke from fires a after nuclear war inspired new research on a concept now known as "nuclear winter"."

Body of the paper:

"Moreover, he was involved in the first studies on the global effects of a thick smoke layer in the atmosphere produced by fires caused by a possible nuclear war . . . "

and

"Paul's work on smoke from fires after a possible nuclear war and the absorption of sunlight by the smoke (Crutzen and Birks, 1982; Birks and Crutzen, 1983) introduced the concept that the use of nuclear weapons would have global impacts that go much beyond the more obvious direct effects. This work inspired substantial research activity – Turco et al. (1983) and Aleksandrov and Stenchikov (1983), soon followed by Robock (1984) and Covey et al. (1984) calculated the surface temperature response to fires after a nuclear war and introduced the term "nuclear winter (see also Robock et al., 2023)"."

*Paul did reopen the debate about geoengineering (now known as "climate intervention"), but only one particular kind. He only discussed solar radiation modification, and not carbon dioxide removal. In fact, he only discussed one proposed scheme, "albedo enhancement by stratospheric sulfur injections".*

We agree and have reformulated the respective papers of the paper.

Abstract:

"He also initiated the reopening of the debate on "geoengineering" – a concept now referred to as "climate intervention". "

We now say in the paper:

". . . Paul has also sparked a new debate by breaking the taboo regarding a possible cooling of the climate by increasing the earth's albedo through stratospheric sulphur injections (Crutzen, 2006)."

Paul also initiated a new discussion on the question of "geoengineering" (now known as "climate intervention") by discussing the role of albedo enhancements caused by stratospheric sulphur injections (Crutzen, 2006)."

and

"These two papers are on the global consequences of a possible nuclear exchange (nuclear winter, Robock et al., 2023) and on climate intervention (Visioni et al., 2023)."

**"Discussion Papers"**

*Lines 149-150: You should also mention the problem that open access to ACP preprints creates. They are assigned a DOI, and I have seen many submitted and even published papers that reference Discussions papers. Sometimes these are*

*rejected papers that should never be referenced, and all the others are not peer-reviewed and have issues that were resolved in the final versions. Just saying how wonderful ACP is while ignoring this problem is disingenuous.*

We agree with Alan Robock that peer reviewed published papers should not be confused with unrefereed preprints. However, this confusion should have been prevented by the two names "Atmospheric Chemistry and Physics" (ACP, for peer reviewed journal articles) and "Atmospheric Chemistry and Physics Discussions" (ACPD, for preprints under discussion). In any case, these two types of publications are different. We also admit, that when the concept of public discussion of submitted manuscripts was introduced, there was initially some confusion in the community about the status of discussion papers (likely because the idea was new at that time).

However, we think that referring to preprints under consideration at ACP is very similar to referring to manuscripts/preprints on arXiv.org and other preprint servers. The only difference being that ACPD papers were under review for the journal, while preprints on arXiv may or may not be under review in a journal. In particular the status of submitted papers/preprints has been further clarified since early 2023 by retiring ACPD and introducing the preprint server EGUsphere for submitted papers for journals like ACP. EGUsphere is a repository for both standalone preprints (much like arXiv.org) and preprints that are under review in ACP. Retiring ACPD was a step taken to partly address concerns like the one expressed in this comment. In particular, the EGUsphere doi makes the preprint status clearer, while the doi also ensures that people's free speech and intellectual ideas are protected and properly archived – a paper on EGUsphere remains a preprint.

Further, the sytem of manuscript reviewing and editorial decisions at ACP was evaluated independently (Bornmann et al., 2010, 2011). There will also be a paper in this special issue (Ervens et al., 2023) where the publishing process in ACP is discussed in much more detail than appropriate and possible here.

Nonetheless, in order to mention the problem that open access to ACP preprints possibly creates, we have introduced the following text in the paper: "Initially, when the concept of public public peer-review and a public discussion of submitted manuscripts was introduced, there was some confusion in the community about the status of discussion papers. Since then, however, it has become clear that discussion papers are preprints similar to the manuscripts posted on other preprint servers like arXiv.org but with the additional feature of undergoing public peer review and discussion. This is also reflected on the web pages of ACP and other interactive open access journals of the European Geosciences Union (EGU)."

There is also some discussion on this subject below, in the "reply to A. R. Ravishankara".

**Reply to Guy Brasseur**

Guy Brasseur stated in his review: *This is an excellent paper that summarizes very well the life and extraordinary achievements of Paul Crutzen. His strong*

*support to the journal "Atmospheric Chemistry and Physics" is also highlighted.*

We thank Guy very much for this encouraging statement.

*For me, the paper does not require any change. It is well written and covers very well the major contributions coming from Paul. Perhaps, a paragraph could be added to highlight the educational role of Paul. A large number of outstanding scientists (you could cite a few of them) had an extraordinary career and their inspiration clearly came from Paul.*

We agree with his point regarding the educational role of Paul and have added the following text to the paper in response:

"Throughout his career, Paul interacted and collaborated with many important, influential scientists. These collaborations are easily noticeable by investigating his list of publications. But perhaps even more impressive is the educational impact he had. There is a very large number of outstanding scientists, who had – and have – an extraordinary career that started with a PhD, a postdoc or an early scientific interaction with Paul. The inspiration for these careers clearly came from Paul and they started at the various institutions at which he was active."

**Reply to A. R. Ravishankara**

**General**

Many of the comments in the review are by Alan Robock and these comments are addressed in the reply to Alan Robock above. There are however some new comments by A. R. Ravishankara in the review (red comments) which are repeated here (in red). We are very grateful for these comments. The reply and the associated changes in the text of the manuscript are given below (in black).

**Comment 1**

(abstract) First versus one of the first. e.g., Hal Johnston and NOx catalyzed strat ozone loss.

We have emphasised the contribution by Hal Johnston now in the following text in the paper:

"The recognition of the strong effect of $NO_x$ on stratospheric ozone had a major impact, since emissions of $NO_x$ caused by a possible fleet of supersonic planes was the first stratospheric ozone depletion issue that was studied (Johnston, 1971; Crutzen, 1972)."

We know that Paul Crutzen (in conversations) made the point that his 1970 contribution should not be forgotten (Crutzen, 1970, although he felt that sometimes it is), so we decided to not change the wording here.

**Comment 2**

(geological institutions): what does this mean?

We agree and have removed the comment on geological institutions.

**Comment 3**

(football) for American audience- Soccer.

We agree and have added a footnote stating "Football is the game referred to in American English as soccer".

**Comment 4**

(tropospheric ozone): This was also done by Chameides and Walker in 1973. The work of Crutzen and C&W were the key studies that led to the realization that tropospheric ozone is mostly produced in the troposphere.

We agree that the work of Chameides and Walker (1973) needs to be mentioned when talking about tropospheric chemistry. In response to the comment we have now expanded and clarified the discussion; the relevant text reads now in the manuscript:

"Tropospheric chemistry, and in particular the chemical production of of ozone in the troposphere were topics of great importance to Paul (Crutzen, 1996; Fishman et al., 2023). The radical OH (Levy, 1971; Crutzen and Zimmermann, 1991; Crutzen, 1996) is responsible for the oxidation of $CH_4$ (and many other compounds emitted into the atmosphere). It was found that in environments containing sufficient NO, the methane oxidation chain could produce ozone in large quantities – large compared to the downward flux of ozone from the stratosphere to the troposphere – (Crutzen, 1973; Chameides and Walker, 1973; Crutzen, 1974). Paul's work and the work of Chameides and Walker (1973) were the key studies that led to the realisation that tropospheric ozone is mostly chemically produced in the troposphere. In the early seventies, very little was known about homogeneous and heterogeneous reactions affecting the methane oxidation chain in the troposphere, so that initial conclusions remained uncertain (Crutzen, 1974). However some years later, together with Jack Fishman and Susan Solomon, Paul presented observational evidence for a strong in situ tropospheric ozone production (Fishman and Crutzen, 1978; Fishman et al., 1979). Later, Paul's work on tropospheric ozone led him to realise the importance of biomass-burning for the chemistry of the atmosphere (Crutzen et al., 1979; Crutzen and Andreae, 1990)."

**Comment 5**

(NOx cycle): this had a major impact since NOx emissions from SSTs was the first strat ozone depletion issue that was studied.

We agree and have extended the text on the SST issue. It reads now:

"The recognition of the strong effect of $NO_x$ on stratospheric ozone had a major impact, since emissions of $NO_x$ caused by a possible fleet of supersonic planes was the first stratospheric ozone depletion issue that was studied (Johnston, 1971; Crutzen, 1972)."

**Comment 6**

(Open review and open access): You may want to qualify this. Faraday Transactions and the Proceedings of the Combustion Institute did this long time ago, but in a Q&A format.

We agree with this comment and in response we have added the following text to the manuscript: 'Scientific discussions have been documented before the invention of ACP; for example in the "Electronic Transactions in Artificial Intelligence" (ETAI) and the "Journal of Interactive Media in Education" (JIME) (Pöschl, 2012, section "Comparison to earlier initiatives with two- or multi-stage open peer review"). Further, the review process in "Faraday Transactions" and in the "Proceedings of the Combustion Institute" is "open", in the sense that these journals have a long history of meetings and their subsequent publication of the discussion. Faraday Discussions collects questions and answers through delegate discussion during meetings (rather than online or through a text forum); a discussion which is then edited and published alongside the articles in each volume. A similar procedure is followed by the Combustion Institute (see Nicovich and Ravishankara, 1982, for an example). The interactive open access process (as we call it today) with a multi-stage public peer review as practised in ACP, however, had not been introduced in scientific publishing prior to the launch of ACP (Pöschl, 2012).'

[revised manuscript text omitted]